# Removing Bias in Multi-modal Classifiers: Regularization by Maximizing Functional Entropies

**Itai Gat**
Technion

**Idan Schwartz**
Technion

**Alexander Schwing**
UIUC

**Tamir Hazan**
Technion

## Abstract

Many recent datasets contain a variety of different data modalities, for instance, image, question, and answer data in visual question answering (VQA). When training deep net classifiers on those multi-modal datasets, the modalities get exploited at different scales, i.e., some modalities can more easily contribute to the classification results than others. This is suboptimal because the classifier is inherently biased towards a subset of the modalities. To alleviate this shortcoming, we propose a novel regularization term based on the functional entropy. Intuitively, this term encourages to balance the contribution of each modality to the classification result. However, regularization with the functional entropy is challenging. To address this, we develop a method based on the log-Sobolev inequality, which bounds the functional entropy with the functional-Fisher-information. Intuitively, this maximizes the amount of information that the modalities contribute. On the two challenging multi-modal datasets VQA-CPv2 and SocialIQ, we obtain state-of-the-art results while more uniformly exploiting the modalities. In addition, we demonstrate the efficacy of our method on Colored MNIST.

## 1   Introduction

Multi-modal data is ubiquitous and commonly used in many real-world applications. For instance, discriminative visual question answering systems take into account the question, the image and a variety of answers. In general, we treat data as multi-modal if it can be partitioned into semantic features, *e.g.*, color and shape can be treated as multi-modal data.

Training of discriminative classifiers on multi-modal datasets like discriminative visual question answering almost always follows the classical machine learning paradigm: use a common loss function like cross-entropy and employ a standard $\ell_2$-norm regularizer (a.k.a. weight decay). The regularizer favors 'simple' classifiers over more complex ones. These classical regularizers are suitable in traditional machine learning settings that predominantly use a single data modality. Unfortunately, because they favor 'simple' models, their use is detrimental when learning from multi-modal data. Simplicity encourages use of information from a single modality, which often ends up biasing the learner. For instance, visual question answering models end up being driven by a language prior rather than visual understanding [1, 2, 3, 4]. *E.g.*, answering 'how many...?' questions with '2' regardless of the question. Another popular example consists of colored images whose label is correlated with their color modality and their shape modality. In these cases, standard learners often focus on the 'simple' color modality and largely ignore the shape modality [5, 6].

To address this issue, we develop a novel regularization term based on the functional entropy. Intuitively, this term encourages to balance the contribution of each modality to classification. To address the computational challenges of computing the functional entropy we develop a method based on the log-Sobolev inequality which bounds the functional entropy with the functional Fisher information.

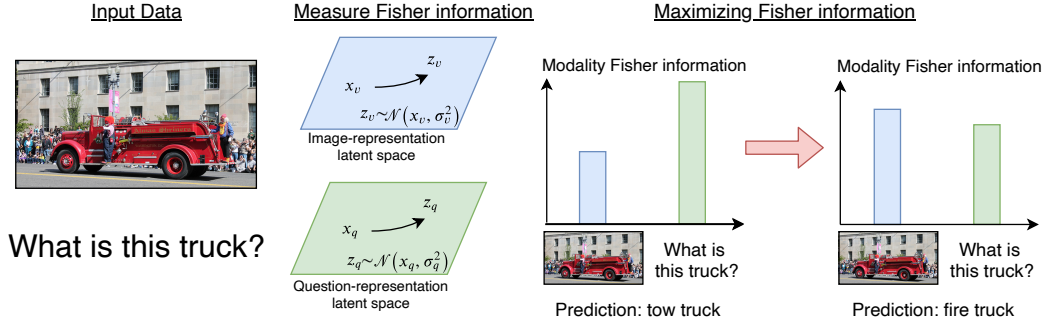

Figure 1: We illustrate our approach. In the visual question answering task, we are given a question about an image. Thus, we can partition our input into two modalities: a textual modality, and a visual modality. We measure the modalities' functional Fisher information by evaluating the sensitivity of the prediction by perturbing each modality. We maximize the functional Fisher information by incorporating it into our loss as a regularization term. Our results show that our regularization permits higher utilization of the visual modality.

We illustrate the efficacy of the proposed approach on the three challenging multi-modal datasets Colored MNIST, VQA-CPv2, and SocialIQ. We find that our regularization maximizes the utilization of essential information. We verify this empirically on the synthetic dataset Colored MNIST. We also evaluate on popular benchmarks, finding that our method permits a state-of-the-art performance on two datasets: SocialIQ (68.53% *vs.* 64.82%) and VQA-CPv2 (54.55% *vs.* 52.05%).

## 2   Related Work

**Multi-modal datasets.** Over the years, the amount and variety of data that has been used across tasks has grown significantly. Unsurprisingly, present-day tasks are increasingly sophisticated and combine multiple data modalities like vision, text, and audio. In particular, in the past few years, many large-scale multi-modal datasets have been proposed [2, 3, 7, 8, 9, 10]. Subsequently, multiple works developed strong models to address these datasets [11, 12, 13, 14, 15, 16, 17, 18, 19, 20, 21, 22, 23, 24, 25, 26]. However, recent work also suggests that many of these advanced models predict by leveraging one of the modalities more than the others, *e.g.*, utilizing question type to determine the answer in VQA problems [1, 27, 28, 29]. This property is undesirable since multi-modal tasks consider all data essential to solve the challenge without overfitting to the dataset.

**Bias in datasets.** Recently, datasets were proposed to study whether a model can generalize and solve the task or whether it uses a single modalities' features. Usually, this evaluation is performed by partitioning data into train and test sets using different distributions. For example, VQA-CP [1] is a reshuffle of the VQA [3] dataset ensuring that question-type distributions differ between train and test splits. Another well-known dataset is Colored MNIST [5, 6, 30]. In this dataset, each digit class is colored differently in the train set, while samples in the test set remain gray-scale. Different approaches were proposed to deal with such problems: Arjovsky *et al*. [30] propose to improve generalization by ensuring that the optimal classifier equals all training distributions. Wang *et al*. [31] suggest to regularize the overfitting behavior to different modalities. Methods like REPAIR [5] prevent a model from exploiting dataset biases by re-sampling the training data. Kim *et al*. [6] use an adversarial approach to learn unbiased feature representations. Clark *et al*. [32] and Cadene *et al*. [33] suggest methods to overcome language priors using a bias-only model in VQA tasks.

**Entropy and information in deep nets.** Entropy plays a pivotal role in machine learning and has been extensively used in losses and for regularization [34]. However, its use is confined to probability distributions while we use functional entropy, which has a different form and is defined for any non-negative function. More broadly, other components of information theory have been studied in deep nets, for example, the information bottleneck criteria [35, 36]. Other works use information theory to overcome generalization [6, 37, 38]. For instance, Krishna *et al*. [37] propose to maximize the mutual information of the modalities by regularizing differences between modality representations. Fisher information is also used in various machine learning and deep learning settings, *e.g.*, monitoring

of the learning process [39]. In contrast, our work considers the *functional* Fisher information of a non-negative function that represents a multi-modal learner, while Fisher information is defined over probability density functions. Also, we use the log-Sobolev inequality between the functional entropy and the functional Fisher information, which does not hold for entropy and Fisher information.

## 3 Background

Discriminative learning constructs mapping between data-instance $x \in \mathcal{X}$ and labels $y \in \mathcal{Y}$ given training data $S$. We are particularly interested in multi-modal data, where each data-instance $x$ is composed of multiple modalities. For example, a monochrome image $x$ is composed of two data modalities $x = (x_c, x_s)$ where $x_c \in \mathbb{R}^3$ is the monochromatic color tone and $x_s \in \mathbb{R}^{d \times d}$ is the $d \times d$ intensity map of the image capturing the shape. Similarly, in discriminative visual question answering, $x$ is composed of a visual modality, a question modality and an answer modality, *i.e.*, $x = (x_v, x_q, x_a)$ with $x_v \in \mathbb{R}^{d_v}$, $x_q \in \mathbb{R}^{d_q}$ and $x_a \in \mathbb{R}^{d_a}$ respectively. Generally, $x$ may have $n$ modalities, *i.e.*, $x = (x_1, \ldots, x_n)$, each residing in Euclidean space, *i.e.*, $x_i \in \mathbb{R}^{d_i}$.

Discriminative learning searches for the parameters $w$ of a function which assign a score to each label $y$ given data $x$. In this work we focus on the softmax function $p_w(\hat{y}|x)$. Its goodness of fit is measured by a loss function, often the cross-entropy loss $\mathrm{CE}(\mathbb{1}[\cdot = y], p_w(\cdot|x)) = -\sum_{\hat{y}} \mathbb{1}[\hat{y} = y] \log p_w(\hat{y}|x)$, where $\mathbb{1}$ refers to the indicator function. More generally, the cross-entropy loss between two distributions $p_w(\hat{y}|x), q(\hat{y})$ is

$$\mathrm{CE}(q, p_w) = -\sum_{\hat{y}} q(\hat{y}) \log p_w(\hat{y}|x). \tag{1}$$

Beyond the loss, a typical learning process employs a regularization term which encourages use of the 'simplest' function. Various regularization terms that favor 'simple' functions pose a considerable difficulty for multi-modal problems: deep learners easily find simple functions that ignore one of the modalities. For example, a simple discriminator for Colored MNIST, which consists of monochromatic images whose colors correlate with their labels, focuses almost exclusively on the color vector to predict the label rather than also assessing the shape of the image. Formally, if the monochromatic images are represented by their color and shape modalities $x = (x_c, x_s)$ then the simplest discriminator will only consider the 3-dimensional color $x_c$. In this setting, the learned function $p_w(\hat{y}|x)$ avoids all important information within the shape modality $x_s$.

In the following we describe the notion of functional entropy in Section 3.1. In Section 3.2 we present the log-Sobolev inequality, which bounds the functional entropy of a non-negative function by the functional Fisher information. We conclude with the notion of tensorization, which decomposes these components according to their multi-modal spaces.

### 3.1 Functional entropy

In this work we consider the functional entropy that is encapsulated in multi-modal problems. Functional entropies are defined over a continuous random variable, *i.e.*, a function $f(z)$ over the Euclidean space $z \in \mathbb{R}^d$ with a probability measure $\mu$. Here and throughout we use $z$ to refer to a stochastic variable, which we integrate over. The functional entropy of a non-negative function $f(z) \geq 0$ is

$$\mathrm{Ent}_\mu(f) \triangleq \int_{\mathbb{R}^d} f(z) \log f(z) d\mu(z) - \left( \int_{\mathbb{R}^d} f(z) d\mu(z) \right) \log \left( \int_{\mathbb{R}^d} f(z) d\mu(z) \right). \tag{2}$$

The functional entropy is non-negative, namely $\mathrm{Ent}_\mu(f) \geq 0$ and equals zero only if $f(z)$ is a constant. This is in contrast to differential entropy of a continuous random variable with probability density function $q(z)$: $h(q) = -\int_{\mathbb{R}^d} q(z) \log q(z) dz$, which is defined for $q(z) \geq 0$ with $\int_{\mathbb{R}^d} q(z) dz = 1$ and may be negative.

### 3.2 Functional Fisher information

Unfortunately, the functional entropy is hard to estimate empirically, since it involves the term $\log(\int_{\mathbb{R}^d} f(z) d\mu(z))$. Since the integral can only be estimated by sampling, the logarithm of its

estimate is hard to compute in practice. Instead of estimating the functional entropy directly, we use the log-Sobolev inequality for Gaussian measures (cf. [40], Section 5.1.1). This permits to bound the functional entropy with the functional Fisher information. Specifically, for any non-negative function $f(z) \geq 0$ we obtain

$$\mathrm{Ent}_\mu(f) \leq \frac{1}{2} \int_{\mathbb{R}^d} \frac{\|\nabla f(z)\|^2}{f(z)} d\mu(z). \tag{3}$$

Hereby, $\|\nabla f(z)\|$ is the $\ell_2$ norm of the gradient of $f$. The functional Fisher information is non-negative, since it is defined for non-negative functions. It is a natural extension of the Fisher information, which is defined for probability density functions.

### 3.3 Tensorization and multi-modal data

Functional entropy naturally fits into multi-modal settings that correspond to product probability spaces. For example, when considering discriminative visual-question answering, a data point $x = (x_v, x_q, x_a)$ resides in the Euclidean product space of the visual modality $x_v$, the question modality $x_q$ and the answer modality $x_a$. This product space property is called tensorization and informally relates the functional entropy of each modality to the overall functional entropy of the system. Generally, consider the product space $\hat{z} = (\hat{z}_1, \ldots, \hat{z}_n)$, where each modality resides in the $d_i$-dimensional Euclidean space $\hat{z}_i \in \mathbb{R}^{d_i}$. Consider the product measure $\mu = \mu_1 \otimes \cdots \otimes \mu_n$ and let

$$f_i(z_i) = f(\hat{z}_1, \ldots, \hat{z}_{i-1}, z_i, \hat{z}_{i+1}, \ldots, \hat{z}_n). \tag{4}$$

The tensorization of the functional entropy amounts to

$$\mathrm{Ent}_\mu(f) \quad \leq \quad \sum_{i=1}^n \int_{\mathbb{R}^d} \mathrm{Ent}_{\mu_i}(f_i) d\mu(\hat{z}), \tag{5}$$

Here the dimension $d$ is the dimension of $\hat{z} = (\hat{z}_1, \ldots, \hat{z}_n)$, namely $\hat{z} \in \mathbb{R}^d$ and $d = \sum_{i=1}^n d_i$. Tensorization is well-suited for multi-modal settings, as it provides the means to bound the overall functional entropy of the system using the functional entropies of its modalities.

## 4 Regularization by Maximizing Functional Entropies

Functional entropy requires a probability measure. In the following we differentiate between multi-modal training points $x = (x_1, \ldots, x_n)$ and general multi-modal points in the probability measure space, which we denote by $z = (z_1, \ldots, z_n)$. We use the training points $x = (x_1, \ldots, x_n)$ to determine the measure and we denote by $z = (z_1, \ldots, z_n)$ the variable of the integrands. In our work we consider a Gaussian product. Given a training point $x \in S$ that resides in the multi-modal space $x = (x_1, \ldots, x_n)$ we define the measure $\mu_i^x$ for the $i$-th modality to be the Gaussian distribution with mean $x_i$ and variance $\sigma_{x_i}^2$, where $x_i$ is the $i$-th modality of the training point $x$ and $\sigma_{x_i}^2$ is the variance of the coordinate of $x_i$:

$$\mu_i^x \triangleq \mathcal{N}(x_i, \sigma_{x_i}^2). \tag{6}$$

The measure $\mu^x$ is the product measure over the different modalities $\mu^x \triangleq \mu_1^x \otimes \cdots \otimes \mu_n^x$. For example, given a monochromatic image $x = (x_c, x_s)$ in the training data, the distribution employed by the functional entropy in Eq. (2) is $\mu^x = \mathcal{N}(x_c, \sigma_{x_c}^2) \otimes \mathcal{N}(x_s, \sigma_{x_s}^2)$.

For each training data point $x \in S$, we define the functional entropy over the deep net softmax function $p_w(\cdot|x)$ as

$$f^x(z_1, \ldots, z_n) \triangleq \mathrm{CE}(p_w(\cdot|z), p_w(\cdot|x)). \tag{7}$$

This function measures the sensitivity of the softmax prediction to Gaussian perturbations $z$ of the input, since the random perturbation $z$ is sampled from a Gaussian with an expected value $x$, as described in Eq. (6).

The cross-entropy function is a non-negative function, therefore, it is natural to apply the log-Sobolev inequality for Gaussian measures to bound the functional entropy using the functional Fisher information, in Eq. (3):

$$\mathrm{Ent}_{\mu^x}\left(\mathrm{CE}(p_w(\cdot|z), p_w(\cdot|x))\right) \leq \int_{\mathbb{R}^d} \frac{\|\nabla_z \mathrm{CE}(p_w(\cdot|z), p_w(\cdot|x))\|^2}{\mathrm{CE}(p_w(\cdot|z), p_w(\cdot|x))} d\mu^x(z). \tag{8}$$

We use the functional Fisher information bound in Eq. (3) to regularize the training process, in order to implicitly encourage to maximize the information of each modality, while minimizing the training loss. In order to account for both the loss minimization and the information maximization, we take the inverse information. Given multi-modal training data $S$, our learning objective is

$$\sum_{(x,y)\in S} \mathrm{CE}(\mathbb{1}[\cdot = y], p_w(\cdot|x)) + \lambda \sum_{(x,y)\in S} \left( \int_{R^d} \frac{\|\nabla_{z^x} \mathrm{CE}(p_w(\cdot|z^x), p_w(\cdot|x))\|^2}{\mathrm{CE}(p_w(\cdot|z^x), p_w(\cdot|x))} d\mu_x(z) \right)^{-1}. \quad (9)$$

The hyperparameter $\lambda$ balances between the training loss and the inverse information.

## 4.1  Tensorization

The tensorization argument in Section 3.3 determines a bound on the functional entropy by its functional entropy over each modality. The tensorization argument is favorable since it permits to consider the functional entropy of each modality separately in the integral of Eq. (5), given a point $\hat{z} = (\hat{z}_1, \ldots, \hat{z}_n)$. The tensorization also permits to efficiently approximate the functional entropy, given a training point $x$: Let $z_i^x \triangleq (x_1, \ldots, x_{i-1}, z_i, x_{i+1}, \ldots, x_n)$ and set $\tilde{f}_i^x(z_i) \triangleq f^x(z_i^x)$. Given this definition, the tensorization in Eq. (5) reduces to

$$\mathrm{Ent}_{\mu^x}(f^x) \quad \leq \quad \sum_{i=1}^n \int_{\mathbb{R}^d} \mathrm{Ent}_{\mu_i^x}(f_i^x) d\mu(\hat{z}) \approx \sum_{i=1}^n \mathrm{Ent}_{\mu_i^x}(\tilde{f}_i^x). \quad (10)$$

We combine this approximation with the log-Sobolev inequality to measure the amount of the functional Fisher information added by each modality, for a given multi-modal training point $x = (x_1, \ldots, x_n)$:

$$\sum_{i=1}^n \mathrm{Ent}_{\mu_i^x} \left( \mathrm{CE}(p_w(\cdot|z_i^x), p_w(\cdot|x)) \right) \leq \sum_{i=1}^n \int_{\mathbb{R}^{d_i}} \frac{\left\| \nabla_{z_i^x} \mathrm{CE}(p_w(\cdot|z_i^x), p_w(\cdot|x)) \right\|^2}{\mathrm{CE}(p_w(\cdot|z_i^x), p_w(\cdot|x))} d\mu_i^x(z_i). \quad (11)$$

We recall that $z_i^x \triangleq (x_1, \ldots, x_{i-1}, z_i, x_{i+1}, \ldots, x_n)$ and $z_i \in \mathbb{R}^{d_i}$ is the variable that is being integrated while all other modalities remain fixed to the training point input modality.

Similarly to Eq. (9), we may use the tensorized functional Fisher information bound in Eq. (11) to regularize the training process. Given multi-modal training data $S$, our tensorized learning objective is

$$\sum_{(x,y)\in S} \mathrm{CE}(\mathbb{1}[\cdot = y], p_w(\cdot|x)) + \lambda \sum_{(x,y)\in S} \sum_{i=1}^n \left( \int_{\mathbb{R}^{d_i}} \frac{\left\| \nabla_{z_i^x} \mathrm{CE}(p_w(\cdot|z_i^x), p_w(\cdot|x)) \right\|^2}{\mathrm{CE}(p_w(\cdot|z_i^x), p_w(\cdot|x))} d\mu_i^x(z_i) \right)^{-1}. \quad (12)$$

# 5  Connection Between Functional Entropy and Variance

Rothaus [41] has shown a connection between the functional entropy of a non-negative function and its variance.

$$\mathrm{Var}_\mu(f) \quad \triangleq \quad \int_{\mathbb{R}^d} f^2(z) d\mu(z) - \left( \int_{\mathbb{R}^d} f(z) d\mu(z) \right)^2. \quad (13)$$

Particularly, when the values of the non-negative function $f(z)$ are small, one can expand the Taylor series of $1 + f(z)$ to show that

$$\mathrm{Ent}_\mu(1 + f) = \mathrm{Var}_\mu(f) + o(\|f\|_\infty^2), \quad (14)$$

where the residual function $o(t)$ is non-negative and approaches zero faster than $t$ approaches zero, *i.e.*, $\lim_{t\to 0} \frac{o(t)}{t} = 0$. Interestingly, a similar bound to the log-Sobolev inequality (Eq. (3)) exists for the variance of continuous random variables $f(z)$, which is widely known as the Poincaré inequality:

$$\mathrm{Var}_\mu(f) \leq \int_{\mathbb{R}^d} \|\nabla f(z)\|^2 d\mu(z). \quad (15)$$

The relation between the functional entropy and the variance, expressed in Eq. (14), suggests that these bounds should behave similarly in practice. To fit the variance into multi-modal settings we

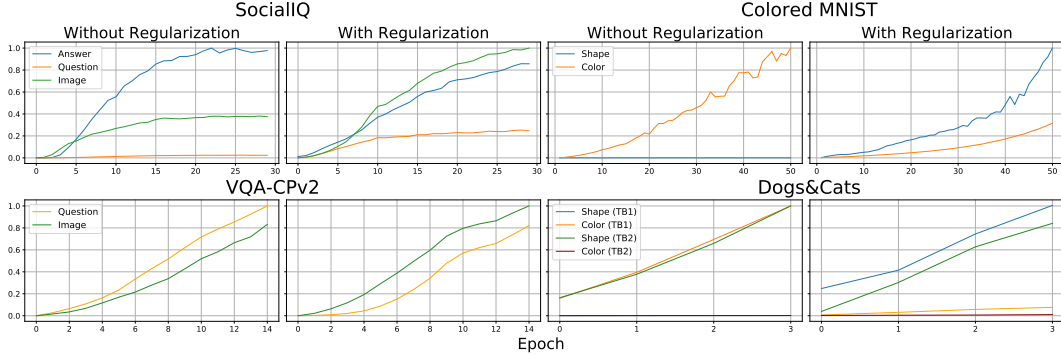

Figure 2: Proportions of the Fisher information values during training for SocialIQ, Colored MNIST, VQA-CPv2 and Dogs&Cats. Using our proposed regularization brings the modalities Fisher information value closer than training without our regularization, a desired property in multi-modal learning. In ColoredMNIST, we observe that training a model with our regularization, the prediction is based on both the shape and the color. Unlike, a model trained without our regularization which makes predictions based on the color only.

need to show tensorization (as in Sec. 3.3). In the variance case, this property is called the Efron-Stein theorem (cf. [42], Proposition 2.2),

$$\text{Var}_\mu(f) \quad \leq \quad \sum_{i=1}^{n} \int_{\mathbb{R}^d} \text{Var}_{\mu_i}(f_i) d\mu(\hat{z}). \tag{16}$$

Next, we present a similar regularization term to the one described in Sec. 4. This time we use variance and Poincaré inequality.

### 5.1 Regularization using Variance

In Sec. 4, we were interested in bounding the functional entropy (Eq. (8)), for each training point $x \in S$, of $\text{CE}(p_w(\cdot|z), p_w(\cdot|x))$. Similarly, we want to bound the variance of $\text{CE}(p_w(\cdot|z), p_w(\cdot|x))$. For this purpose, we can use the Poincaré inequality, described in Eq. (15),

$$\text{Var}_{\mu^x}\left(\text{CE}(p_w(\cdot|z), p_w(\cdot|x))\right) \leq \int_{\mathbb{R}^d} \|\nabla_z \text{CE}(p_w(\cdot|z), p_w(\cdot|x))\|^2 d\mu^x(z). \tag{17}$$

We use the above inequality to regularize the training process. To consider both the loss minimization and the regularization term we formulate the learning objective,

$$\sum_{(x,y)\in S} \text{CE}(\mathbb{1}[\cdot = y], p_w(\cdot|x)) + \lambda \sum_{(x,y)\in S} \left(\int_{\mathbb{R}^d} \|\nabla_{z^x} \text{CE}(p_w(\cdot|z^x), p_w(\cdot|x))\|^2 d\mu_x(z)\right)^{-1}. \tag{18}$$

To fit our multi-modal settings, we need to follow the tensorization process as illustrated in Sec. 4.1. The same tensorization process can be applied to the variance using the Poincaré bound given in Eq. (15). For tensorized Poincaré bound leads to the learning objective

$$\sum_{(x,y)\in S} \text{CE}(\mathbb{1}[\cdot = y], p_w(\cdot|x)) + \lambda \sum_{(x,y)\in S} \sum_{i=1}^{n} \left(\int_{\mathbb{R}^{d_i}} \|\nabla_{z_i^x} \text{CE}(p_w(\cdot|z_i^x), p_w(\cdot|x))\|^2 d\mu_i^x(z_i)\right)^{-1}. \tag{19}$$

## 6 Experiments

In the following, we evaluate our proposed regularization on four different datasets. One of the datasets is a synthetic dataset (Colored MNIST), which permits to study whether a classifier leverages the wrong features. We show that adding the discussed regularization improves the generalization of a given classifier. We briefly describe each dataset and discuss the results of the proposed method.

Table 1: Comparison between our proposed regularization terms on the Colored MNIST (multi-modal settings, gray-scale test set), SocialIQ [7] and Dogs & Cats [6] datasets. We report maximum accuracy observed and accuracy after convergence of the model (Convg). We compare the 4 regularizers specified by the equation numbers. We underline the highest maximum accuracy and bold the highest results after convergence. Using functional Fisher information regularization (Eq. (12)) leads to a smaller difference between the maximum accuracy and accuracy after convergence. * refers to results we achieve without using our proposed regularization. ** denotes training with weight-decay ($\ell_2$ regularization).

| Model | Colored MNIST | | Model | SocialIQ | |
|---|---|---|---|---|---|
| | Convg. | Max | | Convg. | Max |
| Baseline* | 41.11±2.13 | 98.31 | Baseline* | 63.91±0.26 | 66.16 |
| Baseline** | 47.32±1.12 | 98.23 | Baseline** | 65.28±0.23 | 66.95 |
| Eq. (2) | 93.68±0.75 | 94.44 | Eq. (2) | 63.87±0.34 | 64.22 |
| Eq. (13) | 94.87±1.03 | 96.37 | Eq. (13) | 64.36±0.31 | 64.93 |
| Eq. (12) | **96.17**±0.63 | 98.38 | Eq. (12) | **67.93**±0.18 | 68.53 |
| Eq. (19) | **96.24**±0.74 | 98.52 | Eq. (19) | 67.41±0.21 | 68.19 |

(a) Comparison on Colored MNIST.  (b) Comparison on SocialIQ.

| Model | Dogs & Cats (TB1) | | Dogs & Cats (TB2) | |
|---|---|---|---|---|
| | Convg. | Max | Convg. | Max |
| Baseline* | 79.22±0.45 | 80.12 | 65.51±1.54 | 67.38 |
| Baseline** | 81.24±0.23 | 84.31 | 68.47±0.29 | 71.36 |
| Eq. (2) | 92.92±0.46 | 93.48 | 85.32±0.41 | 85.79 |
| Eq. (13) | 93.38±0.27 | 94.15 | 85.14±0.29 | 85.41 |
| Eq. (12) | **94.71**±0.37 | 95.99 | **88.11**±0.17 | 88.48 |
| Eq. (19) | **94.43**±0.24 | 95.35 | 87.81±0.31 | 88.12 |

(c) Comparison on Dogs & Cats.

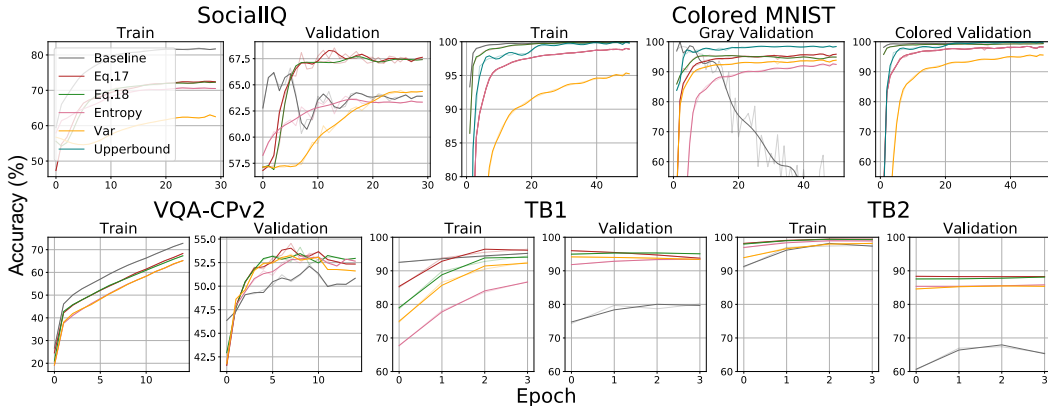

Figure 3: Training process with and without regularization. We note that generalization significantly improves when using our proposed regularization.

## 6.1 Colored MNIST

**Dataset:** Colored MNIST [5, 6] is a synthetic dataset based on MNIST [45]. The train and validation set consist of 60,000 and 10,000 samples, respectively. Each sample is biased with a color that correlates with its digit. The biasing process assigns to each digit an RGB vector which represents a mean color. Then, each sample receives its color, sampled from a normal distribution with a fixed variance around the digit's mean color. This process results in a monochromatic image and high correlation between the digit's color and its label. To introduce a bias in a multi-modal approach, we split each sample $x$ into a color modality $x_c$ and a shape (gray-scale representation of the image) modality $x_s$. For humans it is evident that a digit should be classified based on its shape and not its

Table 2: Comparison between the state-of-the-art on the VQA-CPv2 test set. The best results for each category are in bold. * denotes models that make use of external data.

| Model | Overall | Answer type | | |
|---|---|---|---|---|
| | | Yes/No | Number | Other |
| BUTD [15] | 39.34 | 42.13 | 12.29 | 45.29 |
| AdvReg [38] | 41.17 | 65.49 | 15.48 | 35.48 |
| HINT [43]* | 46.73 | 67.27 | 10.61 | 45.88 |
| RUBi [33] | 47.11 | 68.65 | 20.28 | 43.18 |
| SCR [44]* | 49.45 | 72.36 | 10.93 | **48.02** |
| LMH [32] | 52.05 | 72.58 | 31.12 | 46.97 |
| LMH +Ours Eq. (19) | $54.01 \pm 0.27$ | $73.02 \pm 1.21$ | $43.15 \pm 1.01$ | $47.02 \pm 0.28$ |
| LMH +Ours Eq. (12) | $\mathbf{54.55} \pm 0.29$ | $\mathbf{74.03} \pm 1.13$ | $\mathbf{49.16} \pm 1.22$ | $45.82 \pm 0.37$ |

color. For a learner this fact is not as clear. To minimize the loss, it is much easier for a classifier to leverage the color modality, which correlates very well with the label. In its nature, Colored MNIST evaluates the generalization of a model since it has a test set that assesses whether a classifier relies solely on color or both the color and the shape.

**Baseline:** A simple deep net achieves high accuracy on both colored train and colored validation set. However, on the gray-scale validation set, the network fails drastically, achieving only a 41.11% accuracy when using the model from the last training epoch. We note that the more we train the more the baseline relies on color rather than shape. We also compute an upper-bound by training the deep net on a gray-scale version. The upper-bound accuracy on the gray-scale validation set is 98.47%.

**Results:** Adding our proposed regularization encourages to exploit information from both shape and color modalities. We provide results in Tab. 1. Fig. 2 shows that without entropy regularization, the Fisher information value of the shape is almost zero while adding the regularization results in a higher shape information value than the color. This fact complements the classifier's performance on the gray-scale validation set shown in Fig. 3. Using functional Fisher information based regularization outperforms the same classifier trained without regularization by almost 55%.

## 6.2 VQA-CPv2

**Dataset:** VQA-CPv2 [1] is a re-shuffle of the VQAv2 [46] dataset. Visual question answering (VQA) requires to answer a given question-image pair. [1] observed that the original split of the VQAv2 dataset permits to leverage language priors. To challenge models to not use these priors, the question type distributions of the train and validation set were changed to differ from one another. VQA-CPv2 consist of 438,183 samples in the train set and 219,928 samples in the test set.

**Results:** We evaluated our method by adding functional Fisher information regularization to the current state-of-the-art [32]. In doing so, the result improves by 2.5%, achieving 54.55% accuracy. We provide a comparison with recent state-of-the-art methods in Tab. 2.

The authors of [47, 48] raise the concern that new regularization methods mainly boost the performance of yes/no questions. Investigating the improvements due to our result shows that this is not the case. The accuracy difference to the previous state-of-the-art on the different answer types is: yes/no +1.5%, number +18%, and other -1%.

## 6.3 SocialIQ

**Dataset:** The SocialIQ dataset is designed to develop models for understanding of social situations in videos. Each sample consists of a video clip, a question, and an answer. The task is to predict whether the answer is correct or not given this tuple. The dataset is split into 37,191 training samples, and 5,320 validation set samples. Note that an inherent bias exists in this dataset: specifically the sentiment of the answer provides a good cue.

**Baseline:** A simple classifier based on only the answer modality performs significantly better than chance level accuracy (using our settings ~6% more). Such biases in the train set lead to a classic case of overfitting.

**Results:** As seen in Fig. 3, training without functional Fisher information regularization leads to ~80% accuracy on the train set and ~64% accuracy on the validation set. Although, functional Fisher information regularization results in 70% accuracy on the train set, it improves validation set accuracy to 67.93% accuracy.

We further investigate the information values during the training phase with and without functional Fisher information regularization. In Fig. 2 we observe that without our regularization, the answer modality has the highest information value while the question modality is almost entirely ignored. Adding the proposed regularization balances the information between modalities, the desired behavior in multi-modal learning.

### 6.4 Dogs and Cats

**Dataset:** Following the settings of Kim *et al*. [6], we evaluate our models on the biased "Dogs and Cats" dataset. This dataset comes in two splits: The TB1 set consists of bright dogs and dark cats and contains 10,047 samples. The TB2 set consist of dark dogs and bright cats and contains 6,738 samples. We use the image as a single-modality.

**Baseline:** The authors show that training of ResNet-18 [49] on TB1 and testing on TB2 results in a poor performance of 74.98%. The authors also show that using TB2 as the train set and TB1 as the test set results in even worse accuracy of 66.45%.

Functional Fisher information regularization training on TB1 and testing on TB2 with $\lambda$ (see Eq. (12)) set to equal 3e-10 results in 94.71% accuracy, exceeding [6] by 3.5%. Training on TB2 while testing on TB1 achieves an accuracy of 88.11%, 1% higher than [6].

## 7   Conclusion

Classical regularizers applied on multi-modal datasets lead to models which may ignore one or more of the modalities. This is sub-optimal as we expect all modalities to contribute to classification. To alleviate this concern we study regularization via the functional entropy. It encourages the model to more uniformly exploit the available modalities.

**Acknowledgements:** This work is supported in part by NSF under Grant # 1718221, 2008387, NIFA award 2020-67021-32799 and, BSF under Grant# 2019783.

## Broader Impact

We study functional entropy based regularizers which enable classifiers to more uniformly benefit from available dataset modalities in multi-modal tasks. We think the proposed method will help to reduce biases that present-day classifiers exploit when being trained on data which contains modalities, some of which are easier to leverage than others.

We think this research will have positive societal implications. With machine learning being used more widely, bias from various modalities has become ubiquitous. Minority groups are disadvantaged by present-day AI algorithms, which work very well for the average person but are not suitable for other groups. We provide two examples next:

1. It is widely believed that criminal risk scores are biased against minorities[1], and mathematical methods that reduce the bias in machine learning are desperately needed. In our work we show how our regularization allows to reduce the color modality effect in colored MNIST, which hopefully facilitates to reduce bias in deep nets.

2. Consider virtual assistants as another example: if pronunciation is not mainstream, replies of AI systems are less helpful. Consequently, current AI ignores parts of society.

To conclude, we think the proposed research is a first step towards machine learning becoming more inclusive.

## Footnotes

[1] `https://www.propublica.org/article/bias-in-criminal-risk-scores-is-mathematically-inevitable-researchers-say`

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
