[Supplementary Material]

# Supplementary Material: Removing Bias in Multi-modal Classifiers: Regularization by Maximizing Functional Entropies

**Itai Gat**[*]
Technion

**Idan Schwartz**
Technion

**Alexander Schwing**
UIUC

**Tamir Hazan**
Technion

In this supplementary material we study additional quantitative (see Sec. 1) and qualitative (see Sec. 2) results. We detail our experimental settings (see Sec. 3). Finally, we provide the code we use to perform the experiments (see Sec. 4).

## 1 Quantitative Evaluation

To study robustness, in Tab. S1, we provide analysis for an ensemble of different models on VQA-CPv2. For each model, we used different values of $\lambda$, and different regularization, *i.e.*, Eq. (17) and Eq. (18). Using our Top-3 ensemble, we improved the accuracy by ~2% compared to our single model (56.74% *vs*. 54.55%), and ~4% compared to the previous single model baseline (56.74% *vs*. 52.05%).

In Tab. S2, we show a per question type comparison on VQA-CPv2. Our method improved significantly the "how many people are" question type that requires counting (19.98% *vs*. 42.81%). However, for the "is there" question type which requires image detection, our method's performance dropped in a 19.6% accuracy (22.4% *vs*. 42%).

Table S1: A comparison of different ensemble models on VQA-CPv2. We underline the best single-model and bold the best ensemble model.

| Model | Overall | Answer type | | |
|---|---|---|---|---|
| | | Yes/No | Number | Other |
| Eq.(17) | 54.01 | 73.02 | 43.15 | 47.02 |
| Eq.(18) | 54.55 | 74.03 | 49.16 | 45.82 |
| Top-3 | **56.74** | 77.70 | 48.56 | **48.01** |
| Top-4 | 56.41 | 77.34 | 48.05 | 47.74 |
| Top-5 | 56.54 | **77.48** | **48.72** | 47.71 |

Table S2: Comparison between VQA-CPv2 models in the paper and an ensemble of three models. We show the differences between the state-of-the-art [1] and our method in parenthesis.

| Question-type | LMH [1] | Eq.(18) | Eq.(17) | Ensemble |
|---|---|---|---|---|
| are there | 35.4 | 28.89 (-6.51) | 32.22 (-3.18) | 32.86 (-2.54) |
| what brand | 26.75 | 27.35 (0.6) | 21.84 (-4.91) | 22.76 (-3.99) |
| what room is | 92.57 | 93.22 (0.65) | 92.66 (0.09) | 93.95 (1.38) |
| what color is | 77.18 | 75.53 (-1.65) | 78.03 (0.85) | 78.35 (1.17) |
| is | 77.21 | 69.78 (-7.43) | 78.74 (1.53) | 75.13 (-2.08) |
| are they | 66.78 | 62.0 (-4.78) | 61.67 (-5.11) | 67.33 (0.55) |

---

[*]itaigat@technion.ac.il

| | | | | |
|---|---|---|---|---|
| what number is | 3.28 | 4.22 (0.94) | 3.07 (-0.21) | 2.88 (-0.4) |
| what does the | 17.67 | 17.54 (-0.13) | 17.49 (-0.18) | 18.33 (0.66) |
| is this person | 73.15 | 68.21 (-4.94) | 74.89 (1.74) | 71.87 (-1.28) |
| is the | 36.83 | 40.32 (3.49) | 36.61 (-0.22) | 37.91 (1.08) |
| what is the man | 56.14 | 58.22 (2.08) | 57.21 (1.07) | 58.53 (2.39) |
| what kind of | 42.83 | 45.7 (2.87) | 44.26 (1.43) | 45.75 (2.92) |
| does this | 68.91 | 60.54 (-8.37) | 65.7 (-3.21) | 62.23 (-6.68) |
| is there a | 40.38 | 46.92 (6.54) | 49.62 (9.24) | 52.31 (11.93) |
| is he | 78.14 | 77.3 (-0.84) | 83.11 (4.97) | 78.97 (0.83) |
| what | 34.95 | 36.75 (1.8) | 35.03 (0.08) | 36.19 (1.24) |
| does the | 72.75 | 83.36 (10.61) | 75.84 (3.09) | 81.49 (8.74) |
| is the person | 72.8 | 78.0 (5.2) | 68.96 (-3.84) | 79.24 (6.44) |
| where is the | 26.15 | 28.07 (1.92) | 24.47 (-1.68) | 28.55 (2.4) |
| what animal is | 32.81 | 39.45 (6.64) | 36.83 (4.02) | 34.37 (1.56) |
| how | 17.38 | 19.79 (2.41) | 17.69 (0.31) | 18.82 (1.44) |
| what is the woman | 45.83 | 46.62 (0.79) | 47.0 (1.17) | 47.55 (1.72) |
| what is this | 59.05 | 60.47 (1.42) | 58.51 (-0.54) | 61.55 (2.5) |
| which | 31.68 | 32.79 (1.11) | 30.82 (-0.86) | 32.34 (0.66) |
| where are the | 35.22 | 33.85 (-1.37) | 32.83 (-2.39) | 36.11 (0.89) |
| are the | 44.31 | 42.94 (-1.37) | 44.31 (0.0) | 46.15 (1.84) |
| how many people are | 19.98 | 42.81 (22.83) | 35.98 (16.0) | 37.08 (17.1) |
| what is on the | 35.28 | 36.62 (1.34) | 36.82 (1.54) | 38.37 (3.09) |
| has | 68.72 | 73.53 (4.81) | 77.65 (8.93) | 81.7 (12.98) |
| was | 61.11 | 61.67 (0.56) | 60.28 (-0.83) | 63.89 (2.78) |
| what type of | 47.71 | 49.6 (1.89) | 48.14 (0.43) | 50.38 (2.67) |
| is this an | 58.79 | 48.83 (-9.96) | 53.9 (-4.89) | 46.67 (-12.12) |
| do | 37.75 | 39.25 (1.5) | 42.0 (4.25) | 39.5 (1.75) |
| can you | 64.0 | 81.84 (17.84) | 70.13 (6.13) | 80.9 (16.9) |
| who is | 25.08 | 25.19 (0.11) | 25.49 (0.41) | 25.17 (0.09) |
| are these | 48.59 | 51.77 (3.18) | 50.91 (2.32) | 51.95 (3.36) |
| do you | 68.97 | 82.41 (13.44) | 71.36 (2.39) | 81.88 (12.91) |
| what time | 20.36 | 24.13 (3.77) | 20.04 (-0.32) | 23.25 (2.89) |
| is the woman | 73.26 | 76.16 (2.9) | 84.11 (10.85) | 80.82 (7.56) |
| is this a | 85.34 | 90.17 (4.83) | 87.38 (2.04) | 92.25 (6.91) |
| what are the | 44.86 | 45.01 (0.15) | 45.27 (0.41) | 46.26 (1.4) |
| what color are the | 67.6 | 64.67 (-2.93) | 67.35 (-0.25) | 68.83 (1.23) |
| why | 18.0 | 17.98 (-0.02) | 18.37 (0.37) | 18.83 (0.83) |
| none of the above | 49.5 | 51.22 (1.72) | 50.64 (1.14) | 57.1 (7.6) |
| what is the person | 61.41 | 62.08 (0.67) | 61.52 (0.11) | 63.96 (2.55) |
| how many people are in | 48.16 | 52.78 (4.62) | 54.3 (6.14) | 58.36 (10.2) |
| is this | 81.59 | 84.46 (2.87) | 82.0 (0.41) | 88.46 (6.87) |
| why is the | 16.78 | 16.92 (0.14) | 17.7 (0.92) | 17.63 (0.85) |
| what is the color of the | 78.81 | 75.87 (-2.94) | 75.86 (-2.95) | 80.08 (1.27) |
| what is | 35.54 | 37.64 (2.1) | 36.58 (1.04) | 39.03 (3.49) |
| what are | 54.15 | 56.32 (2.17) | 54.03 (-0.12) | 55.57 (1.42) |
| is that a | 58.76 | 54.87 (-3.89) | 55.75 (-3.01) | 58.85 (0.09) |
| what is in the | 36.98 | 38.77 (1.79) | 39.35 (2.37) | 40.76 (3.78) |
| what sport is | 89.17 | 91.95 (2.78) | 91.02 (1.85) | 91.64 (2.47) |
| how many | 51.27 | 49.08 (-2.19) | 57.49 (6.22) | 56.32 (5.05) |
| what is the | 41.55 | 42.8 (1.25) | 41.18 (-0.37) | 44.11 (2.56) |
| is it | 68.75 | 71.49 (2.74) | 66.26 (-2.49) | 69.75 (1.0) |
| is the man | 53.54 | 50.3 (-3.24) | 52.13 (-1.41) | 52.56 (-0.98) |
| what is the name | 14.76 | 15.82 (1.06) | 14.37 (-0.39) | 15.95 (1.19) |
| is there | 42.0 | 22.4 (-19.6) | 37.2 (-4.8) | 36.0 (-6.0) |
| what color is the | 63.27 | 64.85 (1.58) | 64.29 (1.02) | 67.89 (4.62) |
| what color | 70.19 | 70.5 (0.31) | 70.44 (0.25) | 73.08 (2.89) |
| are | 45.37 | 45.83 (0.46) | 59.44 (14.07) | 58.58 (13.21) |
| **Overall** | 52.05 | 54.01 (1.96) | 54.55 (2.5) | **56.74 (4.96)** |

## 2 Qualitative Evaluation

In this section, we qualitatively study success and failure cases using the VQA-CPv2 dataset. In Fig. 2 of the main paper, we show the proportions of information in each modality. We note that our regularization permits higher utilization of the visual modality. At the end of the document, we present success (see Fig. S1-S16) and failure cases (see Fig. S17-S32).

## 3 Experimental Settings

In the following, for each experiment, we describe its settings.

### 3.1 Colored MNIST

Recent works introduce different variants to Colored MNIST [2, 3, 4]. The main difference is the way bias is incorporated into the MNIST dataset. For instance, in [2] the authors use $\sigma = 0.1$ and evaluate their method on the gray-scale test set while in [4] the authors use $\sigma = 0.02$ and evaluate on a uniformly-colored test set. We follow the experimental settings introduced in REPAIR [2].

**Model:** We use LeNet [5] to encode the shape, then we concatenate the color representation to the last linear layer of LeNet and pass it through another linear layer with a bias term for classification. We use cross-entropy loss.

**Hyper-parameters considered:** We consider $\lambda$ in the range of $1e-9$ to $1e-11$. We report our results with $\lambda = 1e-10$. Note, the values of the regularization term (*i.e.*, the functional Fisher information) are approximately $1e9$. Therefore, the $\lambda$ hyperparameter acts as a normalizer that scales down the bias term to the $[0, 10^2]$ range.

**Computing infrastructure used:** We use a single RTX2080Ti GPU. The average runtime for each epoch is one minute. The model converges in 35 epochs.

### 3.2 VQA-CPv2

Visual question answering (VQA) requires to answer a given question-image pair. VQA-CPv2 [6] is a re-shuffle of VQAv2 [7] that alleviates dataset priors.

**Model:** We add our regularization to the current state-of-the-art [1]. The current state-of-the-art model is based on "Bottom-Up and Top-Down Attention for Image Captioning and Visual Question Answering" [8].

**Hyper-parameters considered:** We consider $\lambda$ in the range of $1e-10$ to $1e-20$. We report our results with $\lambda = 4e-17$. Note, the values of the regularization term (*i.e.*, the functional Fisher information) are approximately $1e15$. Therefore, the $\lambda$ hyperparameter acts as a normalizer that scales down the bias term to the $[0, 10^2]$ range.

**Computing infrastructure used:** We use two RTX2080Ti GPUs. The average runtime for each epoch is 5 minutes. The model converges in 15 epochs.

### 3.3 SocialIQ

SocialIQ [9] input data is constructed from a tuple of a video, a question about the social situation in the video, and an answer to that question. Tiven the tuple, the task is to predict whether the answer is correct or not.

**Model:** The textual inputs are encoded with BERT [10]. The visual input is encoded using a VGG16 [11]. From BERT we extract the last hidden layer's representation. From VGG16, we extract the layer before the final fully-connected representation of four frames from the video (picked uniformly over the video). Then, we forward all representations through a linear layer. Subsequently, the concatenation of the representations is passed through an MLP consisting of two linear layers for classification. We use binary cross-entropy loss. We use a ReLU [12] activation between all layers and dropout [13] with a dropout rate of 0.2.

**Hyper-parameters considered:** We consider $\lambda$ in the range of $1e{-}9$ to $1e{-}11$. We report our results with $\lambda = 3e{-}10$. Note, the values of the regularization term (*i.e.*, the functional Fisher information) are approximately $1e9$. Therefore, the $\lambda$ hyperparameter acts as a normalizer that scales down the bias term to the $[0, 10^2]$ range.

**Computing infrastructure used:** For the baseline, we use a single RTX2080Ti GPU, the average runtime for each epoch is one minute. The model converges after 20 epochs.

### 3.4 Dogs and Cats

In this task, we classify whether a given image shows a dog or a cat. This task was first presented as a Kaggle competition[2]. The authors of "learning not to learn" [4] introduce a modified version. They collected one split to hold bright dogs and dark cats (TB1) and another split that contains the opposite, *i.e.*, dark dogs and bright cats.

**Model:** We follow the setting of Kim *et al*. [4]: The authors use a ResNet-18 [14] that was pre-trained on ImageNet [15]. We fine-tuning the last fully-connected layer.

**Hyper-parameters considered:** In this dataset, we consider $\lambda$ in the range of $1e{-}10$ to $1e{-}14$. We report our results with $\lambda = 2e{-}10$. Note, the values of the regularization term (*i.e.*, the functional Fisher information) are approximately $1e9$. Therefore, the $\lambda$ hyperparameter acts as a normalizer that scales down the bias term to the $[0, 10^2]$ range.

**Computing infrastructure used:** For the baseline, we use five RTX2080Ti GPUs. The average runtime for each epoch is 3 minutes. The model converges after 5 epochs.

## 4 Code

We use the PyTorch [16] framework to conduct all of our experiments. We also provide a Python package which we wrote to calculate the regularization term. Further explanations of how to integrate our regularization to any multi-modal problem are in the README file.

## Question type: what brand

Question: What brand is the racquet?

GT: wilson, Ours: wilson

Question: What brand of racket is the tennis player using?

GT: wilson, Ours: wilson

Question: What brand is his tennis racket?

GT: wilson, Ours: wilson

## Question type: what room is

Question: What room is this?

GT: bathroom, Ours: bathroom

Question: What room is this?

GT: dining, Ours: dining room

Question: What room is this?

GT: bathroom, Ours: bathroom

## Question type: what is the man

Question: What is the man pushing?

GT: bicycle, Ours: bike

Question: What is the man playing with?

GT: frisbee, Ours: frisbee

Question: What is the man flying in the air?

GT: nothing, Ours: nothing

## Question type: is

Question: Is anyone watching the baseball game?

GT: no, Ours: yes

Question: Is someone having breakfast in bed?

GT: no, Ours: no

Question: IS that glass on the counter?

GT: no, Ours: no

Figure S1

Question type: are they

Question: Are they in the
front or back yard?

GT: front, Ours: front

Question: Are they on a
couch or a bed?

GT: bed, Ours: bed

Question: Are they on grass
or dirt?

GT: dirt, Ours: dirt

Question type: what number is

Question: What number is the
small hand on?

GT: 12, Ours: 12

Question: What number is on
the sidewalk?

GT: no number, Ours: 0

Question type: what sport is

Question: What sport is being
played?

GT: frisbee, Ours: frisbee

Question: What sport is this?

GT: baseball, Ours: baseball

Question: What sport is this
man playing?

GT: baseball, Ours: baseball

Question type: is this person

Question: Is this person homeless?

GT: no, Ours: no

Question: Is this person wearing
appropriate protective gear for skateboarding?

GT: no, Ours: no

Question: Is this person an
adult?

GT: no, Ours: no

Question type: is the

Question: Is the horse one
or multiple colors?

GT: 1, Ours: 1

Question: Is the adult a
male of female sheep?

GT: female, Ours: female

Question: Is the child standing
or sitting?

GT: sitting, Ours: sitting

Question type: what is the name

Question: What is the name
of the animal?

GT: horse, Ours: horse

Question: What is the name
of the clothing type she is
wearing?

GT: dress, Ours: dress

Question: What is the name
of the piece of furniture the
stuffed animal is sitting on?
GT: chair, Ours: chair

Question type: how many

Question: How many giraffe are
on the grass?

GT: 5, Ours: 5

Question: How many lids are
here?

GT: 1, Ours: 1

Question: How many giraffes have
their head down?

GT: 1, Ours: 1

Question type: does this

Question: Does this appear to
be a noisy environment?

GT: no, Ours: no

Question: Does this appear to
be in the United States?

GT: no, Ours: no

Question: Does this appear to
be a noisy environment?

GT: no, Ours: no

## Question type: is there a

Question: Is there a man
or woman on the train?

GT: man, Ours: man

Question: Is there a male
or female on the right side?

GT: male, Ours: male

Question: Is there a man
or woman standing directly behind the
stop sign?
GT: man, Ours: man

## Question type: is that a

Question: Is that a cat
or a dog?

GT: dog, Ours: dog

Question: Is that a gas
or electric stove?

GT: electric, Ours: electric

Question: Is that a fork
or a spoon?

GT: fork, Ours: fork

## Question type: can you

Question: Can you rent surfboards
on this beach?

GT: yes, Ours: yes

Question: Can you see grass?

GT: yes, Ours: yes

Question: Can you turn left
at this intersection?

GT: yes, Ours: yes

## Question type: what

Question: What room of a
house would you find all of
these  items?
GT: kitchen, Ours: kitchen

Question: What season is it?

GT: winter, Ours: winter

Question: What structure is in
the background?

GT: fence, Ours: fence

## Question type: does the

Question: Does the man have
a beard?

GT: yes, Ours: yes

Question: Does the person have
any facial hair?

GT: yes, Ours: yes

Question: Does the fruit appear
to be ripe?

GT: yes, Ours: yes

## Question type: is the person

Question: Is the person blurry?

GT: yes, Ours: yes

Question: Is the person a
good surfer?

GT: yes, Ours: yes

Question: Is the person flipping
a skateboard?

GT: yes, Ours: yes

## Question type: do you

Question: Do you think the
bird is beautiful?

GT: yes, Ours: yes

Question: Do you get milk
from these cows?

GT: yes, Ours: yes

Question: Do you see any
phones?

GT: yes, Ours: yes

## Question type: where is the

Question: Where is the woman
and dog?

GT: beach, Ours: beach

Question: Where is the television?

GT: wall, Ours: on wall

Question: Where is the fridge?

GT: kitchen, Ours: kitchen

Question type: what animal is

Question: What animal is on
the ground?

GT: human, Ours: human

Question: What animal is this?

GT: human, Ours: human

Question: What animal is walking
behind the people?

GT: horses, Ours: horse

Question type: how

Question: How will the man
get back to shore?

GT: boat, Ours: boat

Question: How is the lighting?

GT: bright, Ours: bright

Question: How is the sky?

GT: cloudy, Ours: cloudy

Question type: is this a

Question: Is this a double
decker bus?

GT: yes, Ours: yes

Question: Is this a bar?

GT: yes, Ours: yes

Question: Is this a cake?

GT: yes, Ours: yes

Question type: none of the above

Question: Did the wave get
larger after this image was taken?

GT: yes, Ours: yes

Question: Would you consider the
cyclist in yellow an individual?

GT: yes, Ours: yes

Question: Does each glass have
a different kind of wine?

GT: yes, Ours: yes

## Question type: which

Question: Which sport is this?

GT: tennis, Ours: tennis

Question: Which room is it?

GT: kitchen, Ours: kitchen

Question: Which game are they playing?

GT: baseball, Ours: baseball

## Question type: where are the

Question: Where are the motorcycles parked?

GT: street, Ours: street

Question: Where are the giraffes?

GT: field, Ours: field

Question: Where are the giraffes?

GT: zoo, Ours: inside

## Question type: are the

Question: Are the buildings on the other side of the water near or far?

GT: far, Ours: far

Question: Are the people adults or children?

GT: adults, Ours: adults

Question: Are the men at the top or bottom of the mountain?

GT: bottom, Ours: bottom

## Question type: how many people are

Question: How many people are standing up?

GT: 2, Ours: 2

Question: How many people are wearing glasses?

GT: 2, Ours: 2

Question: How many people are on the bike?

GT: 2, Ours: 2

## Question type: what is on the

Question: What is on the ground?

GT: snow, Ours: snow

Question: What is on the table?

GT: tv, Ours: tv

Question: What is on the ground?

GT: snow, Ours: snow

## Question type: has

Question: Has this photo been enhanced?

GT: yes, Ours: yes

Question: Has the food been cooked already?

GT: yes, Ours: yes

Question: Has any of the pizza shown been eaten?

GT: yes, Ours: yes

## Question type: was

Question: Was this a summer gathering or winter gathering?

GT: summer, Ours: summer

Question: Was this me at home or in a restaurant?

GT: home, Ours: home

Question: Was the photo taken on a farm or in the city?

GT: farm, Ours: farm

## Question type: what type of

Question: What type of gate is this?

GT: wooden, Ours: wooden

Question: What type of building is the couple standing in front of?

GT: church, Ours: church

Question: What type of room is the girl in?

GT: living room, Ours: living room

## Question type: do

Question: Do the trees make
this look like fall or summer?

GT: fall, Ours: fall

Question: Do the objects appear
to be for sale or display?

GT: display, Ours: display

Question: Do he have on
pants or shorts?

GT: shorts, Ours: shorts

## Question type: what is the person

Question: What is the person
squatting called?

GT: catcher, Ours: catcher

Question: What is the person
skiing on?

GT: snow, Ours: snow

Question: What is the person
holding?

GT: umbrella, Ours: umbrella

## Question type: who is

Question: Who is the maker
of this car?

GT: ford, Ours: ford

Question: Who is on the
bed?

GT: cat, Ours: cat

Question: Who is joining the
woman in her 'selfie'?

GT: cat, Ours: cat

## Question type: are these

Question: Are these drum sticks
or chopsticks?

GT: chopsticks, Ours: chopsticks

Question: Are these zebra dead
or alive?

GT: alive, Ours: alive

Question: Are these men or
women?

GT: women, Ours: women

Question type: what is the woman

Question: What is the woman
standing on?

GT: skateboard, Ours: skateboard

Question: What is the woman
standing by?

GT: elephant, Ours: elephant

Question: What is the woman
reaching out for?

GT: bird, Ours: bird

Question type: what are

Question: What are they playing?

GT: horse racing, Ours: horse racing

Question: What are these people
riding?

GT: motorcycle, Ours: motorcycles

Question: What are on their
feet?

GT: skis, Ours: skis

Question type: what time

Question: What time of day
is it?

GT: night, Ours: night

Question: What time of season
is it?

GT: fall, Ours: fall

Question: What time of day
is it?

GT: evening, Ours: afternoon

Question type: is the woman

Question: Is the woman happy
to be losing?

GT: no, Ours: no

Question: Is the woman wearing
a lace dress?

GT: no, Ours: no

Question: Is the woman going
up or downhill?

GT: uphill, Ours: uphill

## Question type: what is the

Question: What is the boy holding?

GT: bat, Ours: bat

Question: What is the player's position behind the batter?

GT: catcher, Ours: catcher

Question: What is the girl wearing on her head?

GT: hat, Ours: hat

## Question type: what are the

Question: What are the colors of the chairs?

GT: brown, Ours: brown

Question: What are the military men cutting?

GT: cake, Ours: cake

Question: What are the zebras doing?

GT: standing, Ours: standing

## Question type: what color are the

Question: What color are the walls?

GT: white, Ours: white

Question: What color are the man's pants?

GT: white, Ours: white

Question: What color are the wheels?

GT: green, Ours: green

## Question type: is he

Question: Is he wearing a helmet?

GT: no, Ours: no

Question: Is he playing baseball?

GT: no, Ours: no

Question: Is he holding fishes?

GT: no, Ours: no

Question type: why

Question: Why is this man
slightly squatting?

GT: skateboarding, Ours: skateboarding

Question: Why are they laying
down?

GT: resting, Ours: resting

Question: Why are the Zebras
in the water?

GT: drinking, Ours: drinking

Question type: what is this

Question: What is this animal
bending down for?

GT: food, Ours: food

Question: What is this woman
holding?

GT: frisbee, Ours: frisbee

Question: What is this a
photo of?

GT: food, Ours: food

Question type: how many people are in

Question: How many people are
in the photo?

GT: 1, Ours: 1

Question: How many people are
in the photo?

GT: 3, Ours: 3

Question: How many people are
in the photo?

GT: 1, Ours: 1

Question type: what color

Question: What color of gloves
do they have on?

GT: white, Ours: white

Question: What color pants is
the person wearing?

GT: black, Ours: black

Question: What color ribbon is
in the girls hair?

GT: white, Ours: white

Question type: is this

Question: Is this the last
course of the meal?

GT: yes, Ours: yes

Question: Is this picture in
full color?

GT: yes, Ours: yes

Question: Is this breakfast?

GT: yes, Ours: yes

Question type: why is the

Question: Why is the purpose
of the umbrellas?

GT: shade, Ours: shade

Question: Why is the person
using a knife?

GT: cutting cake, Ours: cutting cake

Question: Why is the fence
there?

GT: safety, Ours: safety

Question type: what is the color of the

Question: What is the color
of the wall?

GT: yellow, Ours: yellow

Question: What is the color
of the woman's shirt?

GT: white, Ours: white

Question: What is the color
of the sky?

GT: blue, Ours: blue

Question type: what is

Question: What is zoomed in
on in the picture?

GT: keyboard, Ours: keyboard

Question: What is keeping the
vehicle from falling?

GT: kickstand, Ours: kickstand

Question: What is covering the
trees in the background?

GT: snow, Ours: snow

Question type: is this an

Question: Is this an Apple
product?

GT: no, Ours: no

Question: Is this an old
structure?

GT: no, Ours: yes

Question: Is this an English
newspaper?

GT: no, Ours: no

Question type: what is in the

Question: What is in the
mirror?

GT: dog, Ours: dog

Question: What is in the
hanging bags?

GT: fruit, Ours: fruit

Question: What is in the
photo?

GT: kitchen, Ours: kitchen

Question type: what does the

Question: What does the woman
hold in her hands?

GT: ski poles, Ours: ski poles

Question: What does the girl
have on her head?

GT: hair, Ours: hair

Question: What does the man
have in his hand?

GT: surfboard, Ours: surfboard

Question type: what kind of

Question: What kind of fence
is between us and the zebra?

GT: wood, Ours: wood

Question: What kind of meat
is here?

GT: chicken, Ours: chicken

Question: What kind of fruit
is in the basket?

GT: bananas, Ours: bananas

Question type: is it

Question: Is it winter or
fall?

GT: winter, Ours: winter

Question: Is it day or
night?

GT: day, Ours: day

Question: Is it daytime or
nighttime?

GT: daytime, Ours: daytime

Question type: is the man

Question: Is the man skiing
or snowboarding?

GT: snowboarding, Ours: snowboarding

Question: Is the man throwing
or catching the frisbee?

GT: catching, Ours: catching

Question: Is the man wearing
shorts or long pants?

GT: shorts, Ours: shorts

Question type: is there

Question: Is there water or
oil on the windows?

GT: water, Ours: water

Question: Is there more pavement
or grass?

GT: grass, Ours: grass

Question: Is there beer or
cola on the door of the
refrigerator?
GT: beer, Ours: beer

Question type: what color is the

Question: What color is the
house?

GT: blue, Ours: blue

Question: WHAT COLOR is the
sky?

GT: blue, Ours: blue

Question: What color is the
train?

GT: blue, Ours: blue

Question type: what color is

Question: What color is this person's jacket?

GT: yellow, Ours: green

Question: What color is her shirt?

GT: white, Ours: white

Question: What color is his uniform?

GT: white, Ours: white

Question type: are

Question: Are all the vehicles automobiles?

GT: no, Ours: no

Question: Are all players playing for the same team?

GT: yes, Ours: no

Question: Are all the children smiling?

GT: yes, Ours: no

Question type: are there

Question: Are there more water glasses or wine glasses?

GT: wine, Ours: wine

Question: Are there more mashed potatoes or broccoli on the plate?

GT: broccoli, Ours: broccoli

Question: Are there more buses or cars in this photo?

GT: cars, Ours: cars

Figure S16

## Footnotes

[2]`https://www.kaggle.com/c/dogs-vs-cats`

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

## Question type: what brand

Question: What brand is the
man's phone?

GT: apple, Ours: motorola

Question: What brand is the
motorcycle?

GT: ducati, Ours: yamaha

Question: What brand of computer
is the partially closed one?

GT: apple, Ours: dell

## Question type: what room is

Question: What room is this?

GT: office, Ours: dining room

Question: What room is this?

GT: store, Ours: living

Question: What room is the
man probably in?

GT: office, Ours: living room

## Question type: what is the man

Question: What is the man
holding?

GT: skis, Ours: ski poles

Question: What is the man
doing to the man in white?

GT: walking, Ours: catching

Question: What is the man
doing?

GT: watching tv, Ours: playing wii

## Question type: is

Question: Is anybody in the
photo facing the camera?

GT: no, Ours: yes

Question: Is everyone wearing a
helmet?

GT: no, Ours: yes

Question: Is today the dogs
birthday?

GT: no, Ours: birthday

Question type: are they

Question: Are they kite surfing or taking pictures?

GT: kitesurfing, Ours: flying kite

Question: Are they going uphill or downhill?

GT: downhill, Ours: uphill

Question: Are they playing singles or doubles tennis?

GT: singles, Ours: doubles

Question type: what number is

Question: What number is on the destination sign on the bus?

GT: 9, Ours: 38

Question: What number is on the back of the red shirt?

GT: 28, Ours: 0

Question: What number is on the bike?

GT: 9, Ours: 4

Question type: what sport is

Question: What sport is being played?

GT: football, Ours: frisbee

Question: What sport is the bat used in?

GT: baseball, Ours: frisbee

Question: What sport is she playing?

GT: baseball, Ours: tennis

Question type: is this person

Question: Is this person really skinny?

GT: no, Ours: yes

Question: Is this person and athlete?

GT: no, Ours: yes

Question: Is this person waterskiing?

GT: no, Ours: yes

## Question type: is the

Question: Is the H20 above
or below this man?

GT: below, Ours: water

Question: Is the H20 above
or below this man?

GT: below, Ours: top

Question: Is the snowboarder facing
towards or away from the camera?

GT: away, Ours: towards

## Question type: what is the name

Question: What is the name
of the hitter?

GT: batter, Ours: baseball

Question: What is the name
of the hat he is wearing?

GT: cowboy, Ours: hat

Question: What is the name
on of any road signs shown?

GT: sergio dukes ave, Ours: main street

## Question type: how many

Question: How many dogs?

GT: 1, Ours: 2

Question: How many vehicles are
on the road?

GT: 0, Ours: 1

Question: How many sprinkles are
on this donut?

GT: 0, Ours: lot

## Question type: does this

Question: Does this young man
have on proper safety equipment for
the sport?
GT: no, Ours: yes

Question: Does this food have
a vegetable?

GT: no, Ours: yes

Question: Does this belong in
a bathroom?

GT: no, Ours: yes

Question type: is there a

Question: Is there a paved
street or a dirt trail in
this photo?
GT: paved, Ours: road

Question: Is there a painting
or a mirror in the frame?

GT: mirror, Ours: picture

Question: Is there a standard
or international outlet installed?

GT: international, Ours: no

Question type: is that a

Question: Is that a limb
or stick?

GT: limb, Ours: rope

Question: Is that a hot
or cold beverage?

GT: hot, Ours: cold

Question: Is that a man
or a statue of a man?

GT: statue, Ours: woman

Question type: can you

Question: Can you see birds?

GT: yes, Ours: no

Question: Can you name an
advertiser on the fence?

GT: hood, Ours: bank of america

Question: Can you see any
stars in the sky?

GT: yes, Ours: no

Question type: what

Question: What shape is the
red sign?

GT: star, Ours: octagon

Question: What country does this
appear to be in?

GT: india, Ours: usa

Question: What breed of dog
are these?

GT: golden retriever, Ours: terrier

## Question type: does the

Question: Does the woman look like she is happy to be on a date?
GT: yes, Ours: no

Question: Does the suitcase match the color of his pants?
GT: yes, Ours: no

Question: Does the room have a patio or balcony?
GT: yes, Ours: table

## Question type: is the person

Question: Is the person in the image a kid or a midget?
GT: kid, Ours: child

Question: Is the person using this kitchen organized?
GT: yes, Ours: no

Question: Is the person on the snow?
GT: yes, Ours: no

## Question type: do you

Question: Do you see a serving knife?
GT: yes, Ours: no

Question: Do you see an ice cream?
GT: yes, Ours: no

Question: Do you have to carry this bag or does it roll?
GT: carry, Ours: luggage

## Question type: where is the

Question: Where is the bus?
GT: parked, Ours: on road

Question: Where is the train?
GT: station, Ours: on tracks

Question: Where is the man and woman riding an elephant?
GT: forest, Ours: outside

Question type: what animal is

Question: What animal is on top of the monitor?

GT: dragon, Ours: cat

Question: What animal is black and white?

GT: panda, Ours: bear

Question: What animal is that?

GT: ram, Ours: sheep

Question type: how

Question: How long has this person been on the bench?

GT: hour, Ours: 1 hour

Question: How fast is the speed limit?

GT: 25, Ours: fast

Question: How old is the elephant?

GT: old, Ours: young

Question type: is this a

Question: Is this a sandwich?

GT: yes, Ours: no

Question: Is this a Nintendo Wii controller?

GT: yes, Ours: wii

Question: Is this a black and white photo?

GT: yes, Ours: black and white

Question type: none of the above

Question: Have the walkways been shoveled?

GT: yes, Ours: snow

Question: At what approximate point on its body do this animal's stripes stop?

GT: stomach, Ours: head

Question: Can both of the planes go the same speed?

GT: yes, Ours: no

## Question type: which

Question: Which would most likely
be eaten last?

GT: cake, Ours: food

Question: Which arm is he
wearing a brace on?

GT: left, Ours: right

Question: Which toilet is hanged
higher?

GT: left, Ours: right

## Question type: where are the

Question: Where are the people
in the picture?

GT: zoo, Ours: outside

Question: Where are the people
playing?

GT: field, Ours: park

Question: Where are the bikes
parked?

GT: street, Ours: parking lot

## Question type: are the

Question: Are the giraffes enclosed
or roaming free?

GT: enclosed, Ours: zoo

Question: Are the dishes clean
or dirty?

GT: dirty, Ours: clean

Question: Are the three signs
pointing right or left?

GT: left, Ours: right

## Question type: how many people are

Question: How many people are
sitting on the elephant?

GT: 2, Ours: 4

Question: How many people are
on the elephant?

GT: 2, Ours: 3

Question: How many people are
watching the player?

GT: 7, Ours: 4

Question type: what is on the

Question: What is on the
suitcase?

GT: jacket, Ours: clothes

Question: What is on the
clock?

GT: sticker, Ours: clock

Question: What is on the
mountains?

GT: snow, Ours: rocks

Question type: has

Question: Has this suitcase been
on a trip in the past?

GT: yes, Ours: no

Question: Has the wine been
opened?

GT: yes, Ours: no

Question: Has he or she
had wine?

GT: yes, Ours: neither

Question type: was

Question: Was the bread grilled
or toasted?

GT: toasted, Ours: fried

Question: Was this picture taken
in the northern or the southern
hemisphere?
GT: northern, Ours: south

Question: Was this taken by
a human or a camera trap?

GT: camera trap, Ours: man

Question type: what type of

Question: What type of weather
is occurring?

GT: rain, Ours: rainy

Question: What type of picture
do you call this?

GT: selfie, Ours: color

Question: What type of footprints
are in the sand?

GT: dog, Ours: brown

Question type: do

Question: Do the bathroom fixtures appear contemporary or outdated?

GT: contemporary, Ours: real

Question: Do the boys look like they are jumping up or falling down?

GT: jumping up, Ours: standing

Question: Do these umbrella's have a solid color or are they multicolored?

GT: multicolored, Ours: striped

Question type: what is the person

Question: What is the person sitting on?

GT: atv, Ours: motorcycle

Question: What is the person wearing?

GT: shorts, Ours: wetsuit

Question: What is the person painted on the side of the van holding in his hands?

GT: umbrella, Ours: graffiti

Question type: who is

Question: Who is wearing a blue shirt?

GT: skateboarder, Ours: girl

Question: Who is in the water?

GT: people, Ours: surfer

Question: Who is the baby elephant hiding from?

GT: no one, Ours: elephant

Question type: are these

Question: Are these ducks or geese?

GT: geese, Ours: sheep

Question: Are these people in a field or on a street?

GT: street, Ours: ground

Question: Are these fruits or vegetables?

GT: vegetables, Ours: fruit

## Question type: what is the woman

Question: What is the woman doing?

GT: brushing hair, Ours: eating

Question: What is the woman wearing on her face?

GT: nothing, Ours: sunglasses

Question: What is the woman doing?

GT: listening to music, Ours: talking

## Question type: what are

Question: What are these people wearing on their faces?

GT: masks, Ours: sunglasses

Question: What are these brushes used for?

GT: brushing teeth, Ours: cutting

Question: What are his eyes doing?

GT: looking down, Ours: eating

## Question type: what time

Question: What time does the clock show?

GT: 10:05, Ours: 1:10

Question: What time is it?

GT: 10:05, Ours: 3:00

Question: What time is it in this scene?

GT: 1:55, Ours: 3:00

## Question type: is the woman

Question: Is the woman barefooted?

GT: no, Ours: yes

Question: Is the woman having sex?

GT: no, Ours: yes

Question: Is the woman a cook?

GT: no, Ours: yes

## Question type: what is the

Question: What is the metal that most likely makes up the top of this tower?
GT: gold, Ours: clock

Question: What is the little girl holding?
GT: bunny, Ours: baby

Question: What is the fence made of?
GT: wood, Ours: metal

## Question type: what are the

Question: What are the words on the screen?
GT: k roberts photography, Ours: unknown

Question: What are the donuts from?
GT: starbucks, Ours: dunkin donuts

Question: What are the people holding?
GT: signs, Ours: nothing

## Question type: what color are the

Question: What color are the tiles on the floor?
GT: white, Ours: gray

Question: What color are the letters on the board?
GT: white, Ours: silver

Question: What color are the players' shoes?
GT: white, Ours: blue

## Question type: is he

Question: Is he a beginner surfer?
GT: no, Ours: yes

Question: Is he bunting or swinging?
GT: bunting, Ours: baseball

Question: Is he right handed?
GT: no, Ours: yes

## Question type: why

Question: Why does he seem
suspended in the air?

GT: jumping, Ours: skateboarding

Question: Why is there a
chair in the bathroom?

GT: to sit, Ours: yes

Question: Why did the rider
stop?

GT: rest, Ours: yes

## Question type: what is this

Question: What is this man's
job?

GT: farmer, Ours: police

Question: What is this?

GT: stove, Ours: kitchen

Question: What is this man
waiting for?

GT: food, Ours: bus

## Question type: how many people are in

Question: How many people are
in the room?

GT: 4, Ours: 8

Question: How many people are
in the picture?

GT: 10, Ours: 15

Question: How many people are
in this photo?

GT: 4, Ours: 3

## Question type: what color

Question: What color of shirt
is the man on the elephant
wearing?
GT: white, Ours: pink

Question: What color are this
person's sunglasses?

GT: white, Ours: black

Question: What color stands out
in this picture?

GT: red, Ours: pink

## Question type: is this

Question: Is this inside?

GT: yes, Ours: no

Question: Is this area desert-like
or lush and green?

GT: lush and green, Ours: trees

Question: IS this outside?

GT: yes, Ours: no

## Question type: why is the

Question: Why is the road
ahead closed?

GT: festival, Ours: traffic

Question: Why is the picture
funny?

GT: no, Ours: yes

Question: Why is the cat's
face blurred?

GT: its moving, Ours: tired

## Question type: what is the color of the

Question: What is the color
of the vehicles?

GT: green and yellow, Ours: yellow

Question: What is the color
of the catcher's hat?

GT: red, Ours: blue

Question: What is the color
of the bus?

GT: white, Ours: pink

## Question type: what is

Question: What is clear in
the background?

GT: window, Ours: train

Question: What is below the
mirror?

GT: fireplace, Ours: tv

Question: What is white and
under the desk?

GT: trash can, Ours: desk

Question type: is this an

Question: Is this an affluent
neighborhood?

GT: no, Ours: yes

Question: Is this an adult's
bed?

GT: no, Ours: yes

Question: Is this an aerial
view?

GT: no, Ours: yes

Question type: what is in the

Question: What is in the
picture?

GT: bottle, Ours: vase

Question: What is in the
sky?

GT: nothing, Ours: flowers

Question: What is in the
glass?

GT: oil, Ours: beer

Question type: what does the

Question: What does the under
of the clock read?

GT: melrose, Ours: clock

Question: What does the rack
on top of the vehicle hold?

GT: trash, Ours: nothing

Question: What does the writing
say?

GT: slow, Ours: nothing

Question type: what kind of

Question: What kind of vehicle
is this?

GT: jet ski, Ours: boat

Question: What kind of animals
are in the picture?

GT: horse and cow, Ours: cows

Question: What kind of shorts
is he wearing?

GT: khaki, Ours: jeans

## Question type: is it

Question: Is it warm or cold where this photo was taken?

GT: warm, Ours: cold

Question: Is it cold or warm?

GT: warm, Ours: cold

Question: Is it raining or snowing?

GT: raining, Ours: snowing

## Question type: is the man

Question: Is the man or woman driving?

GT: woman, Ours: man

Question: Is the man standing straight or leaning on the surfboard?

GT: leaning, Ours: neither

Question: Is the man with the brown hair wearing a tight shirt or a loose shirt?
GT: loose, Ours: no

## Question type: is there

Question: Is there an odd number of picnic tables or an even number?
GT: odd, Ours: 2

Question: Is there an even number of donuts or an odd number?
GT: odd, Ours: 0

Question: Is there lots of water or mainly land in the picture?
GT: both, Ours: ocean

## Question type: what color is the

Question: What color is the wall in the picture? White and ____?
GT: red, Ours: white

Question: What color is the woman''s shirt?

GT: gray, Ours: green

Question: What color is the chair?

GT: gray, Ours: brown

Question type: what color is

Question: What color is her hair?

GT: red, Ours: blonde

Question: What color is her hair?

GT: red, Ours: brown

Question: What color is his luggage?

GT: red and black, Ours: red

Question type: are

Question: Are both elbows bent?

GT: yes, Ours: no

Question: Are all the people drinking coffee?

GT: no, Ours: coffee

Question: Are both pizzas the same?

GT: yes, Ours: unknown

Question type: are there

Question: Are there most likely a couple eating together or a group?
GT: couple, Ours: yes

Question: Are there two cats or just one looking in a mirror?
GT: 1, Ours: cat

Question: Are there bushes or trees?

GT: bushes, Ours: trees

Figure S32