[Reviews · NeurIPS 2020]

Review 1

Summary and Contributions: This paper proposes a regularizer which aims to encourage multi-modal discriminative classifiers to balance the contribution of each modality in its predictions. The effectiveness of this regularizer is validated on datasets with multiple modalities: Colored MNIST, VQA-CPv2, Dogs and Cats, SocialIQ. Update (following rebuttal and discussions with other reviewers). After reviewing the rebuttal materials and discussing, I maintain my positive assessment of this work. I appreciate the extensive work that went into the rebuttal to address reviewer concerns - I think this will be a useful method for the multi-modal community.

Strengths: 1. Multi-model classification is becoming an increasingly important topic of research, with numerous applications. The proposed regularizers could see widespread adoption and impact. 2. The empirical evidence presented across multiple datasets suggests that the maximising functional entropies can be highly effective in supporting good generalization.

Weaknesses: 1. It would be useful to have an in-depth explanation of when, (if at all), this kind of regularization does not represent a useful inductive bias. For example, if one modality is a duplicate signal of a subset of another, is the model forced to make use of both redundant signals? If one modality signal is purely random, to what extent is the model performance harmed? 2. As the authors note, a number of regularizers have been developed (such as the l2-norm regularizer on the weights) which may be less appropriate for multi-modal data. It may make the work more convincing if the experiments included a wide array of popular regularizers (e.g. different kinds of weight norms etc.) to validate their claim that directly searching for the "simplest" model is not the best course of action for multimodal inputs.

Correctness: The regularizer is motivated intuitively (with regard to the benefit of maximising functional entropy), and validated empirically.

Clarity: Overall, the paper is fairly well written. I struggled to understand some of the equations on my initial read (due to some missing terms), but there is a helpful list of typos in the supplementary material which addressed my confusion.

Relation to Prior Work: The related work is clearly written.

Reproducibility: Yes

Additional Feedback: 1. It would be nice to see an explicit quantification of the tightness of the bounds for functional entropy (even if this is only possible in a "toy" setting with sampling). This would help to build stronger intuition for the functional form of the regularizers. 2. To further help with intuition, one additional experiment could train a classifier with two input modalities under the setting in which one modality was corrupted by noise. By retraining models on different levels of corruption, it would make clear the effect of the regularizer in balancing the contributions of the two modalities when one is "stronger" than the other. 3. One potentially interesting point of discussion might be to comment on other works that have investigated the challenges of learning effectively from multiple modalities and determined that different modalities generalize and overfit at different rates [1] (Note: just to emphasise, this is an "unweighted" suggestion because the paper was only published after the NeurIPS submission date). [1] Wang, Weiyao, Du Tran, and Matt Feiszli. "What Makes Training Multi-Modal Classification Networks Hard?." Proceedings of the IEEE/CVF Conference on Computer Vision and Pattern Recognition. 2020.


Review 2

Summary and Contributions: The paper addresses bias in multi-modal classifiers. The presented approach is a new regularization term inspired by information theory. The authors propose to use functional entropy, approximated via the log-Sobolev inequality with the functional Fisher information, tensorized over all modalities. Specifically, the goal is to maximize the information contributed by each modality, thus leading to a better balance in usage of all modalities. The experiments are carried out on 4 datasets, Color MNIST, Social-IQ, Dogs & Cats and VQA-CP v2, and show promising results, superior to some prior works. UPDATE The authors' response has addressed majority of my concerns. Besides, I may have missed some of the details given in the appendix (the authors did provide pretty extensive supplemental materials), but the paper should still be self-contained or at least refer to the appendix where necessary! Overall, it seems that the proposed approach is interesting and quite general as it leads to consistent improvements over the baselines/prior works, even if some improvements are modest. I now lean towards accepting this paper.

Strengths: The motivation of this work is clear and the topic is relevant to the NeurIPS community. The authors contribute a new perspective on bias in multi-modal (or rather multi-feature) classifiers inspired by information theory. The proposed method provides an interesting insight into how much each modality is “being used” by a classifier by visualizing the proportion of the Fisher information of each modality (see Figure 2). Quantitative results obtained on 4 datasets show results superior to some prior works.

Weaknesses: The paper’s big issue is comparison to prior work. Specifically, there is little or no empirical comparison to prior methods for classifier debiasing, such as REPAIR [5], HEX [a], etc. - There is no comparison to prior work on the Colored MNIST dataset. - There is no comparison to prior work on the Dogs & Cats dataset. Even the results from [6] are not included. - There is no comparison to prior work on the Social-IQ dataset (not even [7] who proposed the dataset). - For the VQA-CP v2 dataset, there is a more comprehensive comparison to the recent debiasing models (specific to VQA), but no discussion of these methods. Besides, many experimental details (Section 5) are missing or confusing, see below. Finally, the paper lacks insights that could be gained from each experiment. I.e. it would have been interesting to see more analysis (including qualitative examples) of when the classifiers’ behavior is improved, when it may be affected negatively, what classes benefit the most, etc. The included visualizations (Figure 2) are interesting, but no details on how these are obtained are included.

Correctness: The authors have made several typos in the equations (forgetting to include the gradient operator), which were addressed in the supplemental material. I was not familiar with the notions of “functional entropy” and “functional Fisher information” prior to reading this paper. I was not able to find any references to these terms besides this work. I was also not able to access some of the references to theoretical findings mentioned by the authors, e.g. [32], to verify these. This makes it hard for me to entirely judge the correctness of the proposed method.

Clarity: An important issue with this work is in its writing, which lacks *a high-level yet informative* description of the proposed approach. The same summary repeated several times (L7-12, L32-36) fails to inform the reader about the specifics of the proposed regularization term, to get an intuition without diving into all the technical details on Section 3. Also Figure 1 is not very helpful, e.g. it is not obvious how to interpret the bar plots on the right (which I guess illustrate the “proportion” of information?). The authors seem to use the term “modality” in a rather free manner, e.g. referring to digit’s color and shape as two modalities, or interpreting the question and answer as different modalities. Something like “feature” or “input” seems perhaps more appropriate. Table 1 is not properly discussed, leaving many unanswered questions. - What are the baselines Eq (2) and (3)? - The authors advocate for Eq (17) as the preferred method (based on the functional entropy), while Eq (18) is a simpler analogous method based on variance. It appears that Eq (18) performs almost as well or sometimes improves over (17). Do the authors have comments on the benefits/trade-offs of Eq (17) vs. (18)? - No comparison to prior work on debiasing! - Baseline* / ** must be a completely different model for each task, right? Seeing them all in one row is rather confusing; besides, no specifics on each model are given. E.g. for Color MNIST - L213 “a simple deep net”, and no details for Social-IQ or for Dogs&Cats. - There is no discussion on the Max observed accuracy vs. the accuracy after Convergence. Why is there such a big difference for Color MNIST in particular? - The Social-IQ dataset includes two evaluation metrics: A2 (binary accuracy) and A4 (multiple choice accuracy), only A2 was reported. The accuracy of [7] was not reported. Figure 2: would be good to include some details on how these plots are obtained using the notation from the approach section. Figure 3: more details are needed to explain the compared methods (i.e. what are the “Entropy” and “Var”).

Relation to Prior Work: Some relevant recent works on debiasing that were not cited: [a] S. Sagawa, P. W. Koh, T. B. Hashimoto, and P. Liang. Distributionally robust neural networks for group shifts: On the importance of regularization for worst-case generalization. arXiv preprint arXiv:1911.08731, 2019. [b] H. Wang, Z. He, and E. P. Xing. Learning robust representations by projecting superficial statistics out. In International Conference on Learning Representations, 2019. The related work section does not discuss the VQA-specific debiasing work in detail.

Reproducibility: Yes

Additional Feedback: The experiments on the Dogs & Cats dataset were not mentioned until Sec 5, while other datasets were mentioned earlier repeatedly. As the proposed method is rather general it could be potentially combined with any downstream model. It would be e.g. good to see the proposed regularization terms applied to several models (on one of the tasks). L243: “A simple classifier based on only the answer modality performs significantly better than chance level accuracy (using our settings ~6% more).” => is this referring to some findings from prior work? Which one? Would be great to include the specifics. Table 1: The Dogs & Cats citation should be [6], not [5]. Table 2: Why “Baseline [14]”, whose “baseline” is it? Table 2: It is interesting that the “Number” questions in particular improve significantly, some discussion/analysis on this would be valuable. Similarly, why do the “Other” questions lose in performance? Figure 2: From the Social-IQ plots, it appears that even after regularization, the questions are not used much, compared to images and answers (L250-254). Have the authors experimented with upweighting the regularization term further?


Review 3

Summary and Contributions: Authors tackle the problem of de-biasing classifiers to input modalities in multimodal datasets and show significant improvements particularly in VQA. Authors propose a regularization technique that maximizes the functional fisher information of each modality. Authors prove their claims on ColoredMNIST, Dogs and Cats and show improvements on two datasets VQA-CPv2, SocialIQ.

Strengths: Authors prove claims on MNIST, Dogs and Cats through extensive experiments. Authors show strong improvements in VQA domain in VQA-CPv2 and SocialIQ. Authors also pre-trained model with easy to understand and verify code.

Weaknesses: I have 2 main concerns: 1. Comparisons with standard regularization methods and early stopping Based on results in Table 1 and Figure 3, it looks like techniques like early stopping and perform similar in most datasets (and slightly worse in some cases). There are no results which help understand the marginal improvement over standard regularization techniques. 2. How much does this regularization scheme inhibit learning from input modalities. Based on the results split by question types in supplementary material, it seems like when using the proposed regularization scheme, the model is unable to leverage need high-level image information (objects, relations) for questions that need it i.e. ("is this person", "are they", "does this"). The performance on such questions drops significantly. The visual component of the information needed to answer such questions is present in detection features (used in bottom-up top-down model). It would be good to know if (and how) does the regularization scheme inhibit learning (if any). While the Have the authors considering evaluating their method on VQA-Rephrasings [1] to test generalization on the language inputs? [1]: https://facebookresearch.github.io/VQA-Rephrasings/

Correctness: Yes.

Clarity: The paper is very well written. The experiments and ablations well accompany the claims of the author.

Relation to Prior Work: Yes.

Reproducibility: Yes

Additional Feedback: Table 1 -- creating terminology for each type of regularization and using that in the table might aid in understanding.


Review 4

Summary and Contributions: This paper proposes a novel regularization term based on the functional entropy to remove bias in multi-modal problems. The proposed method is based on the log-Sobolev inequality, which bounds the functional entropy with the functional-Fisher-information. The experiments are conducted on several datasets, including SocialIQ, Colored MNIST, VQA-CPv2, and Dogs&Cats.

Strengths: [Experiments on various datasets] Experiments are conducted on different datasets, including SocialIQ, Colored MNIST, VQA-CPv2, and Dogs&Cats. [Novel regularization for removing bias] The proposed regularization, which is based on the functional entropy, is novel to the problem of multi-modal bias reduction.

Weaknesses: [Evaluation metric] It is not clear to me why Convg. and Max are used as evaluations metrics in Table 1. More motivation, reasons and references behind this are expected in the rebuttal. [Results on VQA v2] The results on VQA-CP v2 datasets are reported. However, the results on VQA v2 are not given which are widely used to evaluate whether the proposed method over-correct the bias. [Results on VQA-CP v2] Only the results using Eq.(17) and Eq.(18) on top of LMH are reported on VQA-CP v2, which cannot evaluate whether the proposed approach is model-agnostic. Although the paper claims that the improvement is not due to yes/no questions, the reason why number questions rather than yes/no and other questions achieve significant improvement is not provided. [Typos] There are some typos in Eq. (12)-(18), which are corrected in *Supplementary Material*.

Correctness: Yes.

Clarity: The writting of this paper is not good. Although the theory part is interesting, there are too many equations in the paper and the intuitive explanation on how the theorical equations are applied and related to the multi-modal classification problem. This mismatch makes the paper hard to understand. More intuitive motivation behind the theory is expected.

Relation to Prior Work: Yes.

Reproducibility: Yes

Additional Feedback: [Focus on softmax function] In line 86, this paper claims that it focuses on the softmax function. On VQA-CP v2 dataset, LMH uses a binary cross entropy loss for optimization, which is also widely used in the VQA task. In this case, the softmax function over answer distributions is not used. I wonder whether it would be better if the softmax function is not highlighted. ------ After rebuttal ----- After the discussion with other reviewers, I agree to increase my score although my main concerns on VQA-CP v2 are now well addressed. I hope that the following comments can be helpful for the future or final version. I focuses more on VQA-CP v2 rather than other three datasets. The experiments are conducted on four datasets, Colored MNIST, Dogs & Cats, VQA-CPv2, and SocialIQ. The first two datasets are more like single-modal datasets, while the last two datasets are vision-language datasets. Since VQA-CP v2 is one of the only two multi-modal datasets, and this paper highlights more on the results on VQA-CP v2 (a separated table in the main paper, Sec. 2&3 with 2 tables and 16 figures in the supp), the results and analysis are important to show whether the method works well under the multi-modal scenario. The authors did a good job of supplemental results. The results on VQA v2.0 address one of my concerns. However, my main concern about the improvement on VQA-CP v2 is not well addressed. First, only the results over LMH are reported, and more results on other models, especially RUBi which uses a debiasing strategy similar to LMH, are not provided. The lacked results are expected to show whether the proposed approach is generalizable. Second, my concern about why the boost mainly comes from "number" questions was not addressed. In the paper, the authors claimed that there is a concern that recent works mainly achieve improvement on yes/no questions. However, the authors did not explain why improvement on number questions is more reasonable than yes/no questions. It is self-contradictory that the improvement on yes/no questions raises concerns but the one on number questions doesn't. Although this paper cite [38,39] to support their opinion, recent work (https://arxiv.org/abs/2005.09241) also argues that improvement on number questions can be easily obtained by a simple baseline. For fairness, I would not rely on any of the above references to doubt the improvement. Besides, as shown in Table 2, the STOA method LMH achieves significant improvement on number questions. The proposed method over LMH also achieves significant improvement on number questions. Therefore, I wonder whether the improvement is highly related to the baseline method (i.e., LMH). The additional experiments over other baseline models or the explanation on number questions are acceptable.

[Author Response · NeurIPS 2020]

We thank all reviewers for their feedback. We are happy the reviewers agree that our work is novel, insightful and offers a new perspective on bias in multi-modal problems.

**Regularization and early stopping [R1,R3]:** On the right we present results for early stopping and $\ell_p$ regularization on the SocialIQ dataset. The baseline is described in the appendix. Our regularization performs better than classical regularization and early stopping.

|  | epoch 5 | epoch 10 | epoch 15 | epoch 20 |
|---|---|---|---|---|
| Baseline | **66.39%** | 62.33% | 63.91% | 62.78% |
| Baseline+$\ell_2$ | 66.02% | 65.27% | 64.98% | 65.42% |
| Baseline+$\ell_1$ | 65.15% | 64.24% | 62.44% | 64.02% |
| Baseline+$\ell_\infty$ | 63.23% | 64.13% | 63.01% | 64.58% |
| Baseline+Ours | 66.16% | **68.08%** | **67.51%** | **67.29%** |

**Tab. 1 & baselines [R2,R4]:** In retrospect, Tab. 1 is confusing as the baselines are different for each task. The baselines are described in the appendix, Section 4. The VQA-CPv2 baseline is based on [23]. The SocialIQ baseline follows [7]. The Dogs&Cats baseline is ResNet18. Baseline** corresponds to these baselines augmented with weight-decay ($\ell_2$ regularization). Lastly, max vs. convg is also confusing: we used it to emphasize the inconsistent behavior of ColoredMNIST. We attribute it to the synthetic nature of ColoredMNIST. We'll clarify.

**Prior art [R2,R3,R4]:** We'll add a comparison to REPAIR on our setting for ColoredMNIST: Our de-biasing achieves 96% accuracy, while REPAIR achieves 84.33%. We compared our performance on Dogs&Cats to "learning not to learn" [6], see L263-L265: for TB1 we got 94.71% and for TB2 we got 88.11%. [6] obtains 90.29% for TB1 and 87.26% for TB2. We'll update to the best reported accuracy on SocialIQ [7] which is 64.82%, while our method improves accuracy to 67.93%. VQA-Rephrasing: the LMH [23] baseline obtains an accuracy of 49.23%, while our regularization improves accuracy to 51.18%.

**VQA-CPv2 result interpretation [R2,R3,R4]:** Great suggestion to study the differences of VQA-CPv2 question-type results of different models. We don't think we can conclude that one model is better at leveraging high-level image information than another. E.g., 'does the,' 'is the person,' 'are these,' questions are very similar in spirit to 'does this,' 'is this person,' 'are they,' questions: both triplets require intricate image understanding. We improve results on the former three while accuracy drops on the latter three.

**R1:** *Duplicated, subset and corrupted signals:* Thanks for these suggestions. The relevant plots show that our regularization reduces the amount of information from corrupted signals, while improving accuracy:

Figure 1: Duplicated, subset, noisy (with different noise levels) modalities: Fisher information (y-axis) as a function of epoch (x-axis) with and without regularization. Accuracy is provided in the plot title. Noisy image is a Gaussian noise added to the image modality.

*Bound tightness:* The bound is tight for the exponential function $f(z) = e^{tz}$. Since we are using the CE similarity measure over exponential families (through the softmax), our bound tends to be tight.
*Modalities overfit at different rates (Wang2020):* Thank you for pointing out this interesting work. Different from our work, this work regularizes the overfitting behavior of different modalities. We'll cite and discuss this work.

**R2:** *Functional entropy literature:* We acknowledge, finding [32] is not easy due to Covid19, as access to academic libraries is limited. Relevant definitions are also in `https://arxiv.org/pdf/math/0609050.pdf`, Sec. 6. *Clarity:* We'll fix and clarify these 7 points: **1)** The bar plots show the functional Fisher information values; **2)** Answer and question are considered as a "modality" in many VQA works [11, 13, 15, 16, 17, 18]. We wanted to be consistent with prior work; **3)** The relation between Eq. (17) to Eq. (18) is indicated by Eq. (4); **4)** VQA-CPv2 is inherently about debiasing and we compare our method to 5 different debiasing models on VQA-CPv2 in Tab. 2. We also compare to the debiasing work "learning not to learn" on Dogs&Cats in Sec. 5.4; **5)** We detail the settings of each model in the appendix (Sec. 4); **6)** We obtain Fig. 2 by computing the functional Fisher information using Eq. (16) for each data-point and then average over all data-points. In Fig. 3 we use Eq. (2) for 'Ent' and Eq. (3) for 'Var'; **7)** We'll add the citations. *SocialIQ A2 and A4:* We evaluated A4 with a similar model to A2. Our regularization improves the accuracy in this task as well: we obtain 56.35% accuracy without our regularization, and we get 57.13% with our regularization. *Different models for the same task:* Thanks for suggesting, we ran SCR [25] on VQA-CPv2 with our regularization and obtained an accuracy of 49.4%. Without our regularization, we obtain 48.8%. We'll add more models on VQA-CPv2 for the camera-ready. *Answer modality bias in SocialIQ (L243):* We noticed it while experimenting. We clarify and provide the code. *Upweighting regularization term:* When upweighting $\lambda$ the modalities tend to increase their functional Fisher information at the expense of accuracy. We'll add plots and a discussion.

**R4:** *Results on VQA v2:* Thanks for pointing out. We used LMH [23] with our regularization and obtain an overall accuracy of 57%, 'yes/no': 66.62%, 'number': 37.97% and 'other': 54.74%. LMH accuracy without our regularization is 56.345%, 'yes/no': 65.057%, 'number': 37.631% and 'other': 54.687%. We obtain consistent improvements.
*Focus on softmax function:* as mentioned in L108 the only constraint on $f$ is non-negativity. It can hence be applied to BCE. Note, BCE can be reduced to CE via a binary softmax probability.

[Meta-Review · NeurIPS 2020]

All reviewers recommend accept after the author response & discussion. The reviewers value the paper for its contributions including - novel approach: functional fisher information based regularization for different modalities - extensive experimental evaluation on 4 datasets I agree with this assessment and accept. A major concern with the work is that the authors provide lots of correction/fixes/additions in the appendix & rebuttal and the appendix is rather extensive. This was considered as a reason for rejection, but the reviewers and AC see the strength of the paper and 1) expect that the authors will follow up and revise the paper accordingly. 2) it is important that the authors should aim to make it a self contained paper which does not require a typical reader to read the appendix or prior work for understanding it, but rather clearly refer to the appendix for additional information/results. 3) to clarify early on in the paper what the authors mean by (multi-) modality as this is typically understood as different semantic modalities, e.g. text & images, rather than different "features"/representations of the same modality. 4) If possible the authors should consider including additional results on VQA-CP v2 as suggested by R4.